# Computational Chemistry to Repurposing Drugs for the Control of COVID-19

**Majid Hassanzadeganroudsari [1]**, **Amir Hossein Ahmadi [2]**, **Niloufar Rashidi [3]**, **Md Kamal Hossain [1]**, **Amanda Habib [1]** and **Vasso Apostolopoulos [1,\*]**

[1] Institute for Health and Sport, Victoria University, Melbourne 3011, Australia; majid.hassanzadeganroudsari@vu.edu.au (M.H.); md.hossain18@live.vu.edu.au (M.K.H.); amanda.habib@live.vu.edu.au (A.H.)

[2] Department of Mechanical Engineering, Faculty of Immunology, K. N. Toosi University of Technology, Tehran 19697, Iran; Amir.H.ahmadi@email.kntu.ac.ir

[3] Department of Chemical Engineering, Science and Research Branch, Islamic Azad University, Tehran 1477893855, Iran; nilourashidi@gmail.com

\* Correspondence: Vasso.Apostolopoulos@vu.edu.au

**Abstract:** Thus far, in 2021, 219 countries with over 175 million people have been infected by severe acute respiratory syndrome coronavirus 2 (SARS-CoV-2). SARS-CoV-2 is a positive sense, single-stranded RNA virus, and is the causal agent for coronavirus disease (COVID-19). Due to the urgency of the situation, virtual screening as a computational modeling method offers a fast and effective modality of identifying drugs that may be effective against SARS-CoV-2. There has been an overwhelming abundance of molecular docking against SARS-CoV-2 in the last year. Due to the massive volume of computational studies, this systematic review has been created to evaluate and summarize the findings of existing studies. Herein, we report on computational articles of drugs which target, (1) viral protease, (2) Spike protein-ACE 2 interaction, (3) RNA-dependent RNA polymerase, and (4) other proteins and nonstructural proteins of SARS-CoV-2. Based on the studies presented, there are 55 identified natural or drug compounds with potential anti-viral activity. The next step is to show anti-viral activity in vitro and translation to determine effectiveness into human clinical trials.

**Keywords:** COVID-19; SARS-CoV-2; computational modeling; molecular docking; molecular dynamics; nonstructural proteins; docking score

## 1. Introduction

Coronavirus disease (COVID-19), which is caused by severe acute respiratory syndrome coronavirus 2 (SARS-CoV-2), was reported in Wuhan, Hubei Province, China on 17 November 2019, and was designated as a risk to public health on 30 January 2020 [1–3]. On 11 March 2020, the World Health Organization declared COVID-19 a pandemic [4]. COVID-19 follows an extremely heterogeneous route, where many patients are either asymptomatic or experience moderate signs of illness, while others experience an aggressive and rapid progression of the disease, in some cases leading to collapsed lungs and multi-organ failure [5]. At the time of writing, SARS-CoV-2 has been detected in almost all countries, with over 170 million confirmed cases and over 3.9 million deaths [6]. Thus far, some vaccines have been developed by different institutes, but there are controversies on the side effects, which result in hesitancy in vaccinations, in addition to the limitations in vaccinations between countries; thus, until the deterministic eradication, prolonged physical distancing, strict hygiene measures and expanded hospital capacity are necessary to halt the spread [7].

The Coronaviridae family comprises enveloped, single-stranded RNA and positive sense viruses. In the 1960s, human coronaviruses were reported in patients with the

common cold [8]. Until now, α, β, γ and σ have been identified as Coronaviridae's four genera. Both α coronavirus (NL63 and HCoV-229E) and the β coronavirus (SARS-CoV, HCoV-OC43, HCoV-HKU1 and MERS-CoV), have been detected in humans [9]. Virus genome sequencing results could detect β-CoV strain in all of five patients [10]. The SARS-CoV, MERS-CoV and SARS-CoV-2 phylogenetic tree is presented by Li et al. [11]. The SARS-CoV-2 virus contains four structural proteins: (1) Spike protein (S), (2) Envelope protein (E), (3) Nucleocapsid protein (N) and (4) Membrane protein (M) (Figure 1) [12]. In all types of emerged coronaviruses, the Spike protein appears as a "crown" [13]. Spike proteins act as mediators for the attachment of the virus to the host cell, and the fusion of host cell membranes with virion particles [14]. Both SARS-CoV and SARS-CoV-2 bind to angiotensin-converting enzyme 2 (ACE2) [15,16]. In ACE2–Spike protein fusion, the S protein is cleaved via proteolytic mechanisms into two subunits: S1 and S2 [17,18]. Each structural protein plays a significant role in the survival and life cycle of the virus, and among other proteins, the Spike protein serves major viral functions [19,20] and is the main target for vaccines for generation of neutralizing antibodies [21]. The Coronaviridae family Envelope protein (E) is composed of 76–109 amino acids, which vary from 8.4 to 12 kilodalton (kDa) in size [22]. The E protein is an integral membrane protein, and its structure is formed by hydrophilic terminals comprising 7–12 amino acids, a hydrophobic transmembrane domain of 25 amino acids, followed by a long hydrophilic carboxyl region [23]. The E protein is involved in protein pathogenesis and assembly, which can be migrated by ion channels [24]. One of the essential proteins in viral assembly and transcription is the ribonucleoprotein complex. In fact, in MERS-CoV, the genome is packed in N protein [25]. In SARS-CoV-2, 30,000 nucleotides build the virus genome [10]. During coronavirus infection in the host, of the four structural proteins, N protein is the most immunogenic [26], and so it is frequently targeted by vaccines and drugs. In fact IgM and IgG antibodies have been found against the N protein in recovered COVID-19 patients [27]. The N protein contains two domains: (1) the N-terminal and (2) the C-terminal domains which can bind to viral RNA genomes. In addition, the N protein is highly disordered and positively charged, which gives the opportunity to N protein for binding to nonspecific nucleic protein [26]. In addition to the structural proteins of SARS-CoV-2, there are a number of non-structural proteins such as the main protease (3CLpro, Mpro, Nsp5), Papin Like protease (PLpro, Nsp3) and RNA-dependent RNA polymerase (RdRp), which are necessary for viral replication and survival within the host [28], and as such, are targeted by drugs for anti-viral effects [29].

A report in 2019 demonstrated that the development of a human vaccine requires at least USD 200 million [30]. Given the current critical situation, drug repurposing can be a smart way to fast track drugs for human use whilst discovering new and novel effective therapeutic treatments [31]. In this way, various well-known drugs have already been used in clinical trials, including Azithromycin (an antibiotic) [32], Hydroxychloroquine (an anti-malarial drug) [33] and Remdesivir (an antiviral against Ebola) [34]. However, based on the clinical reports, most of them have disappointing results [35]. In order to alleviate the effects of the dire situation, as well finding a treatment, computational methods are being utilized as they are both cost-effective and fast [36]. Protein 3D structure is imperative for computational drug repurposing. The function of proteins is determined by their structural and chemical properties [37]. Three-dimensional structures of proteins can be predicted by either computational methods such as homology modeling, threading methods and ab initio, or can be evaluated via experimental methods such as protein x-ray crystallography, neutron diffraction, Cryogenic electron microscopy (Cryo-EM) and nuclear magnetic resonance (NMR) [38]. Computational, determination and prediction methods have been developed for protein structure determination, leading to accurate protein and protein–ligand complex structures. Permeation of computational strategies to various aspects of drug repurposing is possible by these advances [39]. As we can see, in comparison between virtual screening (VS), as a computational strategy, and experimental method high-throughput screening (HTS), VS is advantageous due to being a more direct,

effective screening and cost effective [40,41]. VS can be divided into two methods: structure-based and ligand-based. In the case of unknown structural information for target, but known active ligand, the ligand-based method such as quantitative structure activity relationship (QSAR) method can be utilized. The up-side, when active site and structure of the target is available, molecular docking has been used since the early 1980s [42,43]. Molecular docking intends to predict the structure of a ligand within the constraint of a receptor active site, and, calculate the binding affinity or strength of ligand-target [44]. Molecular docking process passes through two interrelated steps: first sampling algorithm, then scoring function.

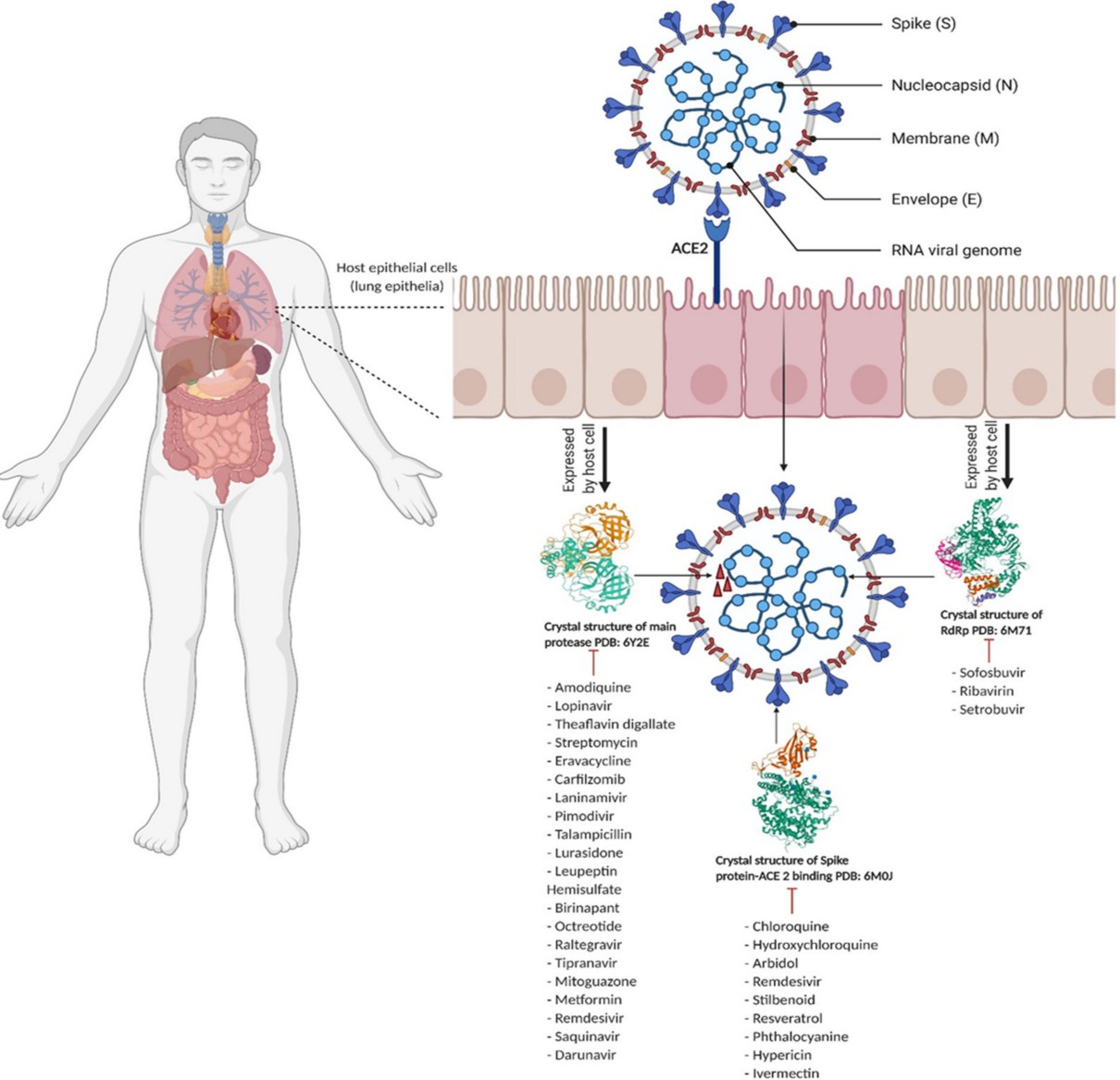

**Figure 1.** Coronavirus entry process showing the Spike protein-ACE2 enzyme binding site and list of drugs that inhibit ACE2 mediated pathway (Created with Biorender.com) [45–47].

Due to numerous conformational degrees of freedom of protein and ligand, it would be an enormous number of binding modes between ligand and structure, and consequently too expensive to computationally afford. Thus, in order to relive the matter, various sampling algorithms with pros and cons have been developed such as: Matching Algorithms (MA) [48], Incremental Construction (IC) [49], Multiple Copy Simultaneous Search

(MCSS) [50], Monte Carlo (MC) [51], Genetic Algorithm (GA) [52] and Molecular Dynamics (MD) [53]. At the next step, the correct poses are delineated from incorrect, as well binded molecules to inactive compounds are omitted by four scoring functions: force-field-based, empirical, knowledge-based and hybrid [54]. From a methodologies stand point, docking can be classified into three methods: rigid ligand and rigid receptor, which is the most simple method with markedly error; flexible ligand and rigid receptor, which is more accurate and widely used; and flexible ligand and flexible receptor docking, which needs high requirements because of a heavy computational burden [55].

Applying computational methods including docking and molecular dynamics, drugs can be repurposed for use in patients with COVID-19 [56]. Ever since the SARS-CoV-2 structure was characterized in February 2020 [57], various computational investigations have been conducted for drug repurposing, targeting both the structural and non-structural proteins of SARS-CoV-2. Due to the huge volume of computational studies published in the last months, we have identified and classified the papers into four subsets: (1) viral protease target, (2) Spike protein–ACE2 enzyme target, (3) RNA-dependent RNA polymerase (RdRp) target and (4) other proteins and non-structural proteins of SARS-CoV-2. In each subset, a summary of the computational studies is presented, identification of potential natural and drug compounds described and an overview of the need for further studies presented. Repurposing drugs already used safely for other diseases is a potential step towards the prevention or treatment of COVID-19 and a potential return to normality worldwide.

## 2. Methodology

### 2.1. Study Selection

Herein, the principles of review preparation [58] regarding the role of computational modeling, specifically virtual screening in drug development against SARS-CoV-2 virus, were followed. This review classifies the various natural and drug compounds docked against different ingredients of SARS-CoV-2 to aid in confusion avoidance, also introducing potential conformers to researchers and allow effective methods and investigations. Research publication databases Scopus, PubMed, Medline, Google Scholar and Embase were searched, up to the 9th of May and including 2021, using appropriate terms related to SARS-CoV-2 computational modeling. All of the identified papers were in the English language and were published in peer-reviewed journals. The terms for screening of the considered field of study contain a term for coronavirus ("SARS-CoV-2" OR "corona" OR "coronavirus" OR "COVID-19" OR "anti-COVID-19" OR "anti-coronavirus"), a term for describing the drug development ("drug" OR "anti-viral" OR "drug development" OR "drug repurposing") and a term for describing computational modeling ("computational modeling" OR "molecular dynamics" OR "molecular docking" OR "virtual screening"). The papers that qualified according to review standards were selected and reviewed. The PRISMA flow diagram and respective information, including the total number of identified papers and those excluded at different stages, is shown in Figure 2.

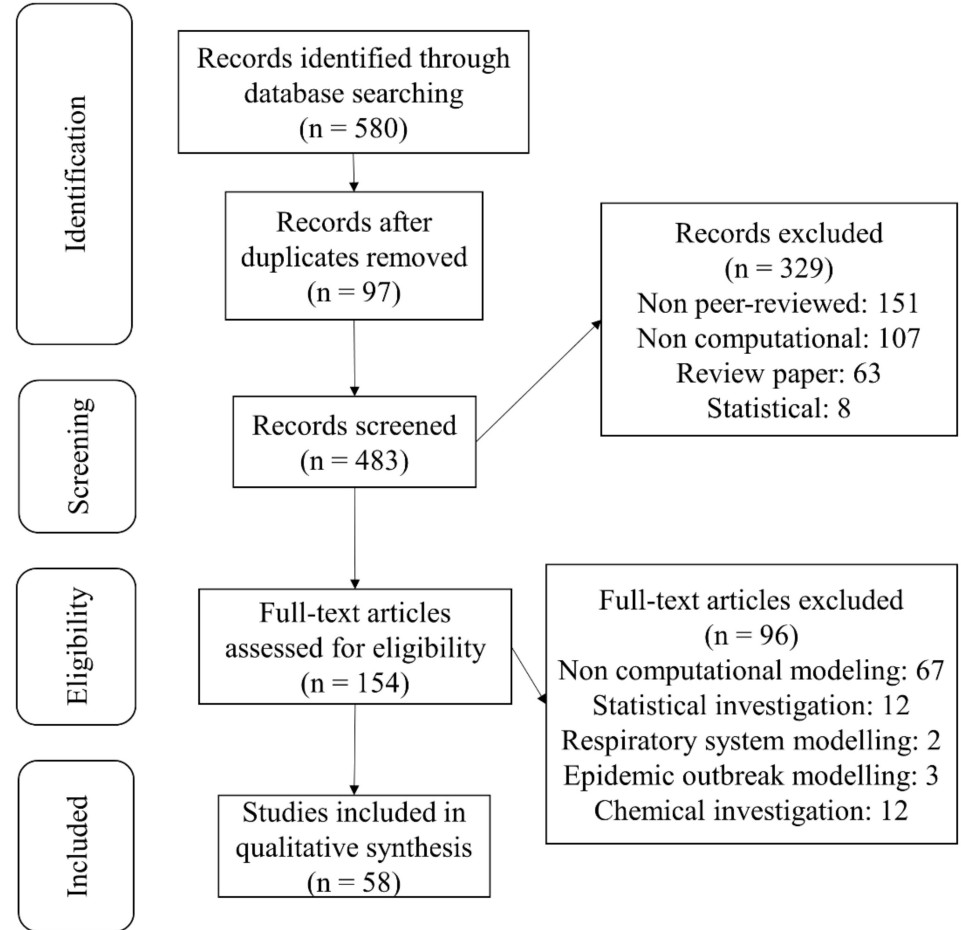

**Figure 2.** PRISMA flow diagram showing the summary of search strategy and paper exclusion. A total of 580 studies were identified by searching databases. By going through the review steps, 58 original articles published in peer-reviewed journals are included regarding computational modeling of SARS-CoV-2.

### 2.2. Data Extraction

Based on the pathogenesis of SARS-CoV-2, there are 6 steps in the viral process: entry, primary replication, spread, secondary replication, secondary viremia and reached target organs [24,59]. The virus at the 'reached target organs' stage disrupts the organs and causes severe complications.

Inclusion criteria included papers that used computational studies with the following targets of SARS-CoV-2, (1) the main protease, (2) the spike protein-ACE2 interface, (3) RNA-dependent RNA polymerase (RdRp) and (4) other proteins and non-structural proteins of SARS-CoV-2. Exclusion criteria at the screening step included: non peer-reviewed, non-computational, review papers and statistical epidemiology. Following the research paper identification step, all abstracts were screened, and 329 papers were excluded at this screening step. After abstract screening at the identification stage, 483 papers were screened as full text, and 154 papers were found to be eligible for selection.

### 3. Computational Studies of Key SARS-CoV-2 Viral Proteins

#### 3.1. Viral Proteases

The crystal structure of main protease and Papain Like protease can provide a basis for computational modeling and drug development [60]. The x-ray structure of main protease of SARS-CoV-2 and also the complex of main protease with an α-ketoamide inhibitor is reported [45]. As the first step, a peptidomimetic α-ketoamides of alphacoronavirus main protease were designed and synthesized [61]. At the next step, by incorporating the P3–P2

amide bond into a pyridine ring, the compound half-life in the plasma is enhanced [45]. The main function of the protease is to break up the polyproteins into non-structural proteins (Nsp) [62]. In the endoplasmic reticulum of the host cell, Nsps assemble and produce membranes as viral RNA synthesis sites [63]. Although the viral replication process is unquestionable [64], their defined role in viral replication is not clear [65,66]. Data on viral proteases can lead to a greater understanding of viral cleavage mechanisms [67], which is why they have been considered for computational modeling studies. Thus far, 33 molecular docking studies on the ligand-proteases docking by different sampling algorithms and scoring functions have been reported (Table 1); and for each study, promising candidate ligands, which are chosen based on the molecular docking and validated docking results by molecular dynamics simulation, are reported.

**Table 1.** Summary of viral protease targets using virtual screening to identify potential repurposed FDA-approved drugs and natural compounds.

| No. | Ligand | Molecular Docking | | | | Molecular Dynamics | | | Ref |
|---|---|---|---|---|---|---|---|---|---|
| - | - | Docking score (Kcal/mol) | Software | Methodology | Binding free energy (Kcal/mol) | Force-field | Simulation time (ns) | RMSD (Å) | |
| 1 | Simeprevier | −11.33 | AutoDuck 4.2 | Not stated | −252.54 ± 85.69 | CHARMM36 | 150 | Not stated | [68] |
| 2 | Lopinavir-Ritonavir | −10.6 | AutoDock Vina | Lamarckian genetic algorithm (GA) in combination with grid-based energy estimation | Not stated | AMBER14 | 10 | 1.5–2.458 | [69] |
| 3 | lopinavir | −9.918 | Schrödinger | HTVS, SP and XP docking modes | Not stated | OPLS | 20 | 2.3 | [70] |
| 4 | carfilzomib | −8.6 | Schrödinger | Glide flexible dockin | −13.8 ± 0.2 | AMBER FF14SB | 125 | Not stated | [71] |
| 5 | PubChem ID: 118098670 | −10.0 | AutoDoc Vina | Hybris scoring function inspired by X-score | Not stated | CHARMM36 | 100 | 3 | [72] |
| 6 | Carvacrol | −4 | AutoDuck 4.2 | Not stated | −19.77 ± 2.24 | GROMOS 96 43a1 | 50 | 2.3–3 | [73] |
| 7 | Leupeptin Hemisulfate | −9.257 | Schrödinger | Not stated | −80.784 | OPLS3 | 200 | Not stated | [74] |
| 8 | Rhizocarpic acid | −9.11 | AutoDock Vina | Xscore as scoring function | −13.81 | CHARMM36 | 10 | 1.7 ± 0.2 | [75] |
| 9 | Pubchem ID: 11610052 | −16.35 | MOE | Not stated | Not stated | GROMOS 96 | 50 | 1.7 ± 0.02 | [76] |
| 10 | Remdesivir | −8.2 | AutoDock 4.2 | Not stated | Not stated | OPLS 2005 | 100 | 1.86 | [77] |
| 11 | Saquinavir | −8.5 | MOE | S-score as scoring function | −36.3026 | AMBER FF14SB | 20 | 2.72 | [78] |
| 12 | Pubchem ID: 129762283 | −9.08 | AutoDock 4.2 | Lamarckian Genetic Algorithm | Not stated | GROMOS 43a1 | 30 | 2.3–2.7 | [79] |
| 13 | Saquinavir | −9.09 | Not stated | HTVS, SP and XP docking modes | −74.4061 | Not stated | 100 | 2.5 | [80] |
| 14 | Gallocatechin-3-gallate | −9 | AutoDock Vina | Not stated | −53.5 | OPLS-AA/L | 100 | 1.45 | [81] |
| 15 | Paritaprevir | −8.8 | AutoDock Vina | MMFF94 Force-field-based | −47.15 | AMBER | 50 | 3.2 | [82] |
| 16 | Conivaptan | −8.6 | AutoDock Vina | Not stated | Not stated | Not stated | 7 | 3.25 | [83] |
| 17 | Indinavir | −8.824 | Schrödinger | XP Gscore | Not stated | Not stated | 100 | 2.771 | [84] |
| 18 | α-ketoamide 13b | −9.2 | Schrödinger | XP scoring function | −25.2 | CHARMM27 | 100 | 2.7 | [85] |
| 19 | Saquinavir | −9.856 | Schrödinger | OPLS_2005 Force-field-based | −72.17 | CHARMM27 | 50 | 0.18 | [86] |
| 20 | PubChem ID: 4167619 | −9.3 | AutoDock Vina | Not stated | −29.3 | CHARMM36 | 20 | 2.2 ± 0.3 | [87] |
| 21 | Salvianolic acid A | −9.7 | Auto Dock | genetic algorithm (GA) | −44.8 | AMBER | 40 | 2.5 | [88] |
| 22 | Desacetylgedunin | −7.3 | Autodock Vina | Not stated | Not stated | CHARMM36 | 40 | Not stated | [89] |
| 23 | ZINC ID: 000621278586 | −9.3 | AutoDock Vina | Not stated | −30.86 ± 0.57 | GROMOS 96 54a7 | 20 | 2.8 ± 0.34 | [90] |
| 24 | Lymecycline | −8.87 | Not stated | Not stated | −22.19 ± 5.23 | AMBER FF14SB | 120 | 3.10 ± 0.43 | [91] |
| 25 | Cadambine | −8.6 | AutoDock | Not stated | −51.92 ± 6.03 | Not stated | 250 | Not stated | [92] |
| 26 | Isoliquiritine apioside | −7.8 | AutoDock Vina 4.2 | Lamarkian genetic algorithm | Not stated | GROMOS 96 43a2 | 100 | 3.41 | [93] |
| 27 | Salicylamide | −7.1 | Schrödinger | HTVS, SP and XP docking modes- glide score as scoring function | −29.042 | AMBER14 | 100 | 1.11 | [94] |

**Table 1.** *Cont.*

| No. | Ligand | Molecular Docking | | | | Molecular Dynamics | | | Ref |
|---|---|---|---|---|---|---|---|---|---|
| 28 | Nelfinavir | −8.3 | AutoDock Vina | Not stated | Not stated | CHARMM36 | 30 | Not stated | [95] |
| 29 | TMC-310911 | −8.3 | AutoDock Vina | Not stated | −52.8 | GAFF2 | 50 | 3.5 | [96] |
| 30 | ABBV-744 | −7.79 | Schrödinger | HTVS, SP and XP docking modes | −45.43 | Not stated | 200 | 2.45 | [97] |
| 31 | Phyllaemblicin C | −9.723 | Schrödinger | Glide molecular docking | Not Stated | GAFF | 60 | Not Stated | [98] |
| 32 | Dpnh (NADH) | −11.016 | Schrödinger | HTVS, SP and XP docking modes | Not Stated | Not Stated | Not Stated | Not Stated | [99] |
| 33 | Saquinavir | −9.5 | Autodock | MMFF94 force-field-based | Not applicable | Not applicable | Not Applicable | Not Appli-cable | [100] |

### 3.2. Spike Protein-ACE2 Enzyme Target

The Spike protein plays a key role in viral entry into host cells and in the pathogenesis of COVID-19. SARS-CoV-2 enters host cells by binding to ACE2 receptors [15,16,19,20]. At first, it was believed that ACE2 was an angiotensin conversion catalyzer in the endothelial cells of the kidney and heart [101,102]. Subsequently, however, the function of ACE2 in binding to SARS-CoV was discovered [103]. ACE2 can transport amino acids, whilst also serving as an important receptor for catalytic activities within the Coronaviridae family [15,104]. There are 380 amino acid differences between SARS-CoV and SARS-CoV-2, of which 5 amino acids are different between Spike protein-ACE2 binding interface [105]. As the Spike protein is regarded as a golden key at the viral entry stage, this protein has become a main target for drug repurposing. Both SARS-CoV-2 and SARS-CoV bind to ACE2 as a cell receptor and their binding mode is almost identical [106,107]. The crystal structure of Spike protein—ACE2 complex was crucial in identifying key amino acids in the interaction [46]. As a result, 13 computational studies have been published in the last three months (Table 2), screening for drugs and compounds, in order to block the ACE2 or the S protein–ACE2 complex.

**Table 2.** Computational studies of spike protein or spike protein–ACE2 enzyme interface targeting of FDA re-purposed drugs and natural compounds.

| No. | Ligand | Molecular Docking | | | Molecular Dynamics | | | | Ref |
|---|---|---|---|---|---|---|---|---|---|
| - | - | Docking score (Kcal/mol) | Software | Methodology | Binding free energy (Kcal/mol) | Force-field | Simulation time (ns) | RMSD (Å) | |
| 1 | Isothymol | −5.7853 | MOE | Rigid protein- flexible ligand | Not stated | Not stated | Not stated | Not stated | [108] |
| 2 | dithymoquinone | −8.6 | Autodock vina | Not stated | −26.7955 | Not stated | 100 | 2.58 | [109] |
| 3 | Resveratrol | −8 | Autodock vina | Not stated | −23.88 | AMBER AFF14SB | 50 | 1.78 | [110] |
| 4 | Orientin | −101.17 | Autodock v4.2 | Not stated | −70.6 | CHARMM | 20 | 4.6 | [111] |
| 5 | phthalocyanine | −16.3 | Autodock Vina | Lamarckian Genetic Algorithm | −66.6 | Not stated | 30 | Not stated | [112] |
| 6 | Theaflavin digallate | −8.7 | AutoDock | Not stated | −38.51 (±1.59) | GROMOS 54a7 | 18 | Not stated | [113] |
| 7 | glycyrrhizic acid | −9.2 | Autodock vina | Not stated | −79.23 | CHARMM36 | 100 | 12.3 | [114] |
| 8 | GR hydrochloride | −11.23 | Autodock vina | Lamarckian Genetic Algorithm | Not Stated | GROMOS 96 43a1 | 50 | 2.5–3.1 | [115] |
| 9 | Lumacaftor | −9.4 | AutoDock | Not stated | Not Stated | Not Stated | Not Stated | 3.2 | [116] |
| 10 | Phyllaemblicin C | −9.131 | Schrödinger | Glide molecular docking | Not Stated | GAFF | 60 | Not Stated | [98] |
| 11 | Coenzyme A | −11.555 | Schrödinger | HTVS, SP and XP docking modes | Not Stated | Not Stated | Not Stated | Not Stated | [99] |
| 12 | Bisoxatin | −7.4 | AutoDock | Not Stated | −31.94 | Not Stated | 100 | Not Stated | [117] |
| 13 | Cefpiramide | −9.1 | Autodock Vina | Not Stated | 19.09 ± 4.45 | CHARMM36 | 10 | Not Stated | [118] |

### 3.3. RNA-Dependent RNA Polymerase Enzyme Target

RNA-dependent RNA polymerase (RdRp) is an enzyme that functions as a catalyzer in the RNA replication process within a virus life cycle [119]. In SARS-CoV-2, the non-structural protein 12 (Nsp12) is known as RdRp, and is associated with the non-structural protein 7 (Nsp7) and non-structural protein 8 (Nsp8), thus playing a key role in SARS-CoV-2 virus replication [120]. The structure of Nsp12 was demonstrated via cryo-electron microscopy [47]. It was noted that Nsp12 in SARS-CoV and SARS-CoV-2 possesses structural homology [121]. Remdesivir, Favipiravir and Ribavirin were shown to be potential drug candidates targeting RdRp protein [122]. Currently, there are five molecular docking against on RdRp and nucleocapsid protein as targets for drug screening (Table 3). Effects of anti-hepatitis C virus inhibitors on RdRp were investigated by Abdo Elifiky [123]. Through molecular dynamic and docking simulations, the binding energy of approved drugs and those against MERS and SARS-CoV were calculated and compared. As a result, Remdesivir, IDX-184, Sofosbuvir and Ribavitin are considered as potent drugs against SARS-CoV-2 [123]. Additionally, by molecular docking simulations, Tenofovir and Golidesivir were added to the list of anti-RdRp drug candidates [124].

**Table 3.** Screening of FDA-repurposed drugs and natural compounds for RNA-dependent RNA polymerase enzyme target.

| No. | Ligand | Molecular Docking | | | | Molecular Dynamics | | | Ref |
|---|---|---|---|---|---|---|---|---|---|
| - | - | Docking score (Kcal/mol) | Software | Methodology | Binding free energy (Kcal/mol) | Force-field | Simulation time (ns) | RMSD (Å) | |
| 1 | IDX-184 | −9 | Autodock Vina | Not Stated | Not Stated | Not Stated | Not Stated | Not Stated | [123] |
| 2 | Sofobuvir | −9.3 | Autodock Vina | Not Stated | Not Stated | Not Stated | Not Stated | Not Stated | [124,125] |
| 3 | CAS ID: 833463-10-8 | −9.529 | Not Stated | Not Stated | Not Stated | CHARMM36 | 20 | 1.6 | [126] |
| 4 | Cryptomisrine | −9.4 | AutoDock Vina | Not Stated | −60.15 | AMBER FF14SB | 4 | 1.87 | [127] |

### 3.4. Other Proteins and Nonstructural Proteins of SARS-CoV-2

In relation to other proteins of SARS-CoV-2 including Nucleocapsid (N) protein, Envelope (E) protein and Membrane (M) protein and nonstructural proteins including Nsp7, Nsp8, Nsp10, Nsp13, Nsp14, Nsp15 and Nsp16, there have been nine computational studies published thus far that pass through our criteria. Among the 16 Nsps, Endoribonuclease (Nsp15) is considered the main viral interferon antagonist [128] and plays an important role in RNA processing [129]. The catalytic site of Nsp15 contains one lysine residue and two histidine residues and is structurally similar with RNase A [130,131]. By computational modelling on Nsp1, 2300 FDA approved drugs were repurposed, and Edoxudine and Remdesivir were identified as potential drugs. In addition, some natural small molecules including Glycyrrhizic acid, Gingeronone and Galangan were found to be promising candidates against Nsp1 [132]. With respect to virtual ligand screening and molecular docking studies of drugs against Nucleocapsid protein [133], 8987 drugs from PubChem and Asinex databases were assessed. According to calculated free binding energies, three drugs were noted to have stability during their interaction with SARS-CoV-2; Zidovudine from PubChem and 6799 and 5817 from Asinex databases. In other studies, 56,079 compounds from the Maybridge library and Asinex database and screened by virtual ligand binding followed by molecular dynamics, two binders, ZINC0000146942 and ZINC00003118440 were identified as inhibitors of the N-terminal domain of the N protein (Table 4) [134]. Additionally, two research studies investigated the Envelope (E) protein of SARS-CoV-2. Gupt et al. [24] studied the effects of phytochemicals on SARS-CoV-2 through the ion channels of E protein. It is concluded that Val25 and Phe26 amino acids can be effected by Vibsanol B and Macaflavanone E. In the Borkotohy study [135], the effects of 70 Indian plant

compounds on membrane and envelope proteins were studied by GROMACS software, based on GROMOS 96-43a1 force field for 200 ns time period.

**Table 4.** Nonstructural proteins of SARS-CoV-2. Summary of nine computational studies.

| Protein/NSP. | Crystal Structure | Ligand | Dock Score | Ref. |
|---|---|---|---|---|
| Nucleocapsid (N) protein |  PDB ID: 6M3M | Asinex ID: 5817 | −10.29 | [133] |
| | | ZINC00003118440 | −6.728 | [134] |
| Envelope (E) protein |  PDB ID: 5X29 | Belachinal | −11.46 | [24] |
| | | Nimbolin A | −11.2 | [135] |
| Membrane (M) protein |  PDB ID: 3I6K | Nimocin | −10.2 | [135] |
| NSP 15 |  PDB ID: 6VWW | Enamine ID: Z595015370 | −10.50 | [136] |
| | | Glisoxepide | −9.4 | [137] |
| | | Pubchem ID: 132519418 | −6.7 | [138] |
| NSP 16 |  PDB ID: 6W75 | Raltegravir | −10.3 | [139] |
| | | Hesperidin | −10.3 | [140] |

## 4. Discussion

The COVID-19 pandemic caused by SARS-CoV-2 virus first became apparent in November 2019 in China [1] and was subsequently declared a global pandemic [2]. Although Rappouli et al. [3] reported that the development of a human vaccine could take

at least 15 years and at least USD 200 million, an alliance of the science society has now achieved this [4]. Besides the hesitation on the authenticity and reported side-effects in a small proportion of people of approved vaccines [141,142], the worldwide vaccination is time consuming, as well as public health being in danger from current virus mutations, which can lead to a prolonged pandemic leading towards a difficult situation [143]. In this rapidly evolving crisis, drug repurposing is considered the quickest way to identify potential drug treatments. Some of the repurposed drugs have already entered into human clinical trials and early data suggests that a number of drugs could be beneficial against SARS-CoV-2. For instance, based on the data analysis of National Institutes of Health, patients who received Remdesivir (a broad spectrum anti-viral), compared to those on placebo, showed significant differences in recovery time [144]. However, according to the report by Wang et al. [145], Remdesivir clinical trials were not found to be beneficial in a Chinese patient cohort. Accordingly, several other therapeutic agents have been developed and repurposed, although none have yet shown to significantly affect SARS-CoV-2 [146]. In addition, Hydroxychloroquine and Chloroquine, anti-malarial drugs, have shown to have anti-SARS-CoV-2 viral activity [147]. However, in a New York medical center clinical study of seventy COVID-19 patients treated with Hydroxychloroquine no beneficial effects were shown, in regard to patient survival [148]. Further, Dexamethasone (a corticosteroid) has also been identified as a potential drug for patients with COVID-19 showing reduced mortality rates in seriously ill patients [149]. However, due to the two properties of dexamethasone, which (1) limits the effect of cytokines [150] and (2) reduces the effects of B cells [151], further research is required.

An abundance of virtual screening models using various molecular docking methods and following that molecular dynamics have been produced. This large volume may overwhelm and confuse researchers. Thus, collection and classification of effective ligands on different ingredients of SARS-CoV-2 was aided by current review. At the first step, several reliable databases such as Google Scholar, Scopus, Embase, PubMed and Medline were investigated for drug repurposing through molecular docking against SARS-CoV-2, and almost all relevant papers published in peer-reviewed journals were collated and analyzed. All the studies, at the first step have identified numerous compounds to bind to SARS-CoV-2 proteins, and those with the lowest docking score (or highest binding affinity) are chosen for further investigation. The compounds are assessed by molecular dynamics to insure whether the conformation is stable, followed by RMSD calculations. In the case that the ligand could pass through the process, it is introduced as a potential compound, which could then be investigated clinically. By searching through search engines, many studies can be achieved. Molecular docking methodologies are various and consequently, diverse parameters are presented. In order to compare the results, we have chosen docking score as the benchmark. The results that had not reported docking scores were eliminated. Finally, 58 Studies are collected and classified in Tables 1–4.

The viral main protease target section consisted of 33 articles. This subset contains the largest number of searches. Most of the studies focused on the identification of repurposed drugs and natural compounds for their ability to bind to the main protease of SARS-CoV-2 for potential anti-viral activity. For molecular docking, various software are utilized. Most studies have not presented the method of docking. Docking scores range from −4 to −16.35. Among 33 studies, Saquinavir is the only ligand presented with the highest binding affinity by four independent reportings [78,80,86,100]. In the first study [78], Saquinavir has interaction with His41 and Cys45 residues. It can form hydrogen bonds with Gly143, Ser144 and Cys145 with −36.3026 (kcal/mol) binding free energy. The second study [80] reports that Saquinavir forms 6 hydrogen binds in Cys145 and His41 as central dyad residues, as well with Gly143, Met49, Gln189 and Glu166, with −74.4061 (kcal/mol) free binding energy. Furthermore, in the third study [86], it was emphasized that Saqinavir interacts with Cys145 with −72.17 (kcal/mol) free binding energy; however, the number of hydrogen bonds was not mentioned. However, in the fourth study, no specific information was reported except docking score [100]. Besides Saquinavir, there are an additional two

drugs with lower docking scores. Qamar et al. [76] reported a markedly lower docking score, −16.35 for Pubchem ID: 11,610,052 with His41, Cys145, Thr24, Thr25, Thr26, Cys44, Thr45, Ser46, Met49, Asn142, Gly143, His164, Glu166 and Gln189 as interacting residues. In addition, Simeprevier [68] with a docking score of −11.33 and free binding energy −252.54 (kcal/mol) also shows promise. The authors claimed that Simeprevier is more effective than Remdesivir, Chloroquine and Hydroxcychloroquine with −5.8, −7.5 and −6.7 docking scores, respectively. Simeprevier forms three hydrogen bonds with His163, Thr26 and Asn119 and interacts with central binding residues Cys145, His41 and Met49.

Regarding the Spike protein-ACE2 enzyme target, 13 original articles are identified and gathered in Table 2. As can be noted, there are a range of docking scores −101.17 for orientin compound with −70.6 (kcal/mol) binding free energy [111]. In this research, orientin is docked to ACE2 receptor and it has been observed that orientin forms five hydrogen bonds with three Lys26, Glu22 and Asn90 residues, as well as interactions with Lys94 and Glu22. The stability of orientin is checked via molecular dynamics simulation, and 4.6 Å is reported on average [111]. The second ligand in prioritization is phthalocyanine with −16.3 docking score and −66.6 (kcal/mol) [112]. It is reported that phthalocyanine establishes 11 interactions with the protein as well as hydrophobic interactions within itself, which increases its stability and complex strength. Moreover, coenzyme A with −11.555 docking score is introduced as a potential drug, but there is no information about the interactions or number of hydrogen binding [99]. Furthermore GR hydrochloride with −11.23 docking score is ranked fourth [115]. It is reported that it forms two hydrogen bonds with ACE2 receptor and interacts with Phe40, Ala348, Trp349, Gly352, Gly354, His378, Asp382, Tyr385, Ala386, Phe390, Arg393, Asn394, and His401 residues. Via computational modeling of RdRp enzyme target, five in silico studies were identified. The presented drugs identified as potential candidates with considerable docking score ranges about −9. In the first study [123], IDX-184 with −9 docking score is reported, which has 18 hydrogen bonds with RdRp and is more stable due to (i) the ability to establish a salt bridge with Asp514 and (ii) it forms two metal interactions with Glu702 and Asp635 residues. In addition, sofosbuvir with −9.3 docking score [124,125] forms eight hydrogen bonds with Asp343, Thr447, Lys442, Ser573, Asp524 and Cys513. Stability of these ligands are not evaluated by molecular dynamics, and thus are faced with strong hesitation. In the fourth study, it is reported that the compound CAS 833463-10-8 with −9.529 docking score can form three hydrogen bonds with Arg553, Tyr455 and Tyr619 [126]. The fifth study has introduced cryptomisirine with −9 docking score that forms 1 hydrogen bond with Thr556 residue [127].

In the other proteins and non-structural proteins of SARS-CoV-2 section, all the miscellaneous computational studies on N protein, M protein, E protein, Nsp7, Nsp8, Nsp10, Nsp13, Nsp14, Nsp15 and Nsp16 are collected and summarized, which can significantly help researchers in their future studies. Among these nine studies, docking score of Belachinal is the lowest with −11.46, which can effect E protein [24]. At the next stage, Nimbolin A with −11.2 docking score and binding free energy of −71.05 (kcal/mol), has been reported to have hydrophobic interactions with Leu18 (A), Leu19 (A), Ala22 (A), Leu19 (B), Leu18 (B), Ala22 (C), Phe23 (C), Phe26 (C), Asn15 (D), Leu18 (D), Leu19 (D), Val25 (D), Phe26 (D), Leu18 (E), Leu19 (E), and Ala22 (E) residues [135]. After collection and classification of various studies, only a limited number of natural or drug ligands have been translated to human clinical studies. At the time of writing the current article, no experimental or clinical studies have been reported on most of these identified natural and drug compounds, but their efficacy testing cannot be ignored. Thus far, Hydroxychloroquine and Chloroquine have been approved by the FDA, but because of disappointing outcomes and modest effects these have undergone controversial debates. A limited number of drugs have been investigated clinically or experimentally investigations such as Remdesivir, Lopinavir/Rotinavir, Sofosbuvir, Carvacol and Favipiravir. Gordon et al. has studied experimentally the effects of Remdesivir on nonstructural proteins of MERS-CoV and has introduced it as a high potency inhibitor against MERS-CoV, which have high structural

similarity with SARS-CoV-2 [152]. Moreover, other research group after 14 days clinical trial of sofosbuvir has reported that duration of hospital stay is significantly reduced for the patients whom have received sofosbuvir [153]. Additionally, sofosbuvir is recommended by some pharmacologists to be considered in clinical trials [154]. Furthermore, it is reported that Lopinavir/ritonavir inhibitor, which is a well-known HIV medication drug, not only has undergone clinical trials [155], but also in some countries it can be used as an alternative drug. In a study on hemodialysis patients, lopinavir/ritonavir is identified as an effective drug through experimental investigation [156]. In addition, in a review paper by Javed et al. that is dedicated to experimental investigation as well clinical trials of carvacrol, it is claimed that due to wide range of experimental studies, carvacrol has protective effects against infective diseases within the scope of SARS-CoV-2 [157]. It should be mentioned that there are some articles about clinical trials of other compounds, but because they are preprints, we have ignored them. One of the important gaps of research is the lack of pharmacokinetic simulation for docked natural or drug compounds, which should be considered for future research; however, the value of current molecular docking and molecular dynamics is not diminished.

### 5. Conclusions

The year 2020 opened with an emerging threat to world health—the coronavirus outbreak. Genomic research suggested that SARS-CoV-2 was transmitted from bats, possibly via pangolins, to humans. Due to the complex nature of COVID-19 epidemiology and virology, there are still many unanswered questions. Discovering therapeutic anti-viral agents is currently the main focus due to the prevalence and rapid spread of the virus, as well as its relatively high death rate. In this critical situation, computational modeling, specifically virtual screening, offers the fastest, cheapest, and most effective method for discovering effective drugs. Based on the studies presented, there are almost 55 identified natural and drug compounds with potential anti-viral activity. However, much research is still required for their effectiveness in human clinical trials.

**Author Contributions:** Conceptualization, M.H. and A.H.A.; methodology, M.H.; investigation, N.R., M.K.H. and A.H.; writing—original draft preparation, M.H., A.H.A. and V.A.; writing—review and editing, V.A.; supervision, V.A.; All authors have read and agreed to the published version of the manuscript.

**Funding:** This research received no external funding.

**Data Availability Statement:** No new data were created or analyzed in this study. Data sharing is not applicable to this article.

**Acknowledgments:** V.A. would like to thank the place-based planetary health grant PH098 for their support. V.A. would also like to thank The Pappas Family, the Greek Orthodox Archdiocese of Australia and the Victoria University vaccine appeal, whose generous philanthropic support made possible the preparation of this paper. The authors would also like to thank the Immunology and Translational Research Group for their significant contribution. The Mechanisms and Interventions in Health and Disease Program within the Institute for Health and Sport, Victoria University Australia are also appreciated for their support. M.K.H. and A.H. were supported by Victoria University Postgraduate Scholarships and M.K.H. was also supported by the Vice-Chancellors top-up Scholarship Award.

**Conflicts of Interest:** The authors declare no conflict of interest.

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
