# Peer review of "Computational Chemistry to Repurposing Drugs for the Control of COVID-19"

_biologics, doi:10.3390/biologics1020007_

Round 1

Reviewer 1 Report

Some other publications are highly related to this review paper are suggested to be included:

  1. M.A. Khazeei Tabari et al. Applying computer simulations in battling with COVID-19, using pre-analyzed molecular and chemical data to face the pandemic. Informatics in Medicine Unlocked 21 (2020) 100458, https://doi.org/10.1016/j.imu.2020.100458
  2. Qiao, Z.; Zhang, H.; Ji, H.-F.; Chen, Q. Computational View toward the Inhibition of SARS-CoV-2 Spike Glycoprotein and the 3CL Protease. Computation 20208, 53. https://doi.org/10.3390/computation8020053

Overall, the revised version illustrated very insight analysis to show that molecular modeling being a strong tool in repurposing drugs. This is a very good review paper to learn from different aspects of utilizing different force field and method to investigate the binding affinity of small molecules with receptors.

Author Response

Reviewer 1:

Some other publications are highly related to this review paper are suggested to be included:

  1. M.A. Khazeei Tabari et al. Applying computer simulations in battling with COVID-19, using pre-analyzed molecular and chemical data to face the pandemic. Informatics in Medicine Unlocked 21 (2020) 100458, https://doi.org/10.1016/j.imu.2020.100458
  2. Qiao, Z.; Zhang, H.; Ji, H.-F.; Chen, Q. Computational View toward the Inhibition of SARS-CoV-2 Spike Glycoprotein and the 3CL Protease. Computation 2020, 8, 53. https://doi.org/10.3390/computation8020053

Response: Thank you for your comment. The suggested addition of these publications are now included. All changes has been highlighted in turquoise for reviewer’s consideration. Because the first paper was review paper, we couldn’t bring it into the PRISMA diagram, but we included it in the introduction section (Ref 36). The second paper is included in Table 1 under section 3.1 (Ref 100).

Reviewer 2 Report

The manuscript deals with an important topic which is very relevant for the present time. The authors used computational methodology and data analysis on a number of known drugs to discover new application against Covid-19 as the repurposing approach towards drug discovery. Computational studies used here are primarily molecular docking and virtual screening of compound databases.

However, none of the 55 identified natural and drug compounds by them were actually tested against Covid-19 virus to validate their model. 

Thus, the manuscript still remains a pure theoretical study. Should the authors provide some evidence for anti-Covid 19 activity of a few of the 55 identified compounds, it may be accepted for publication. However, I leave the choice to the editor for final decision. 

Author Response

Reviewer 2:

Should the authors provide some evidence for anti-Covid 19 activity of a few of the 55 identified compounds, it may be accepted for publication.

Response: Thank you for your comment. We have identified clinical trials and experimental use on some of the compounds which show anti-COVID-19 effects. The requested correction has been applied. All changes are highlighted in blue for reviewer’s consideration (Lines 366-386).

Round 2

Reviewer 2 Report

The authors have not shown any evidence on their own identified compounds. They have shown evidence on remdesivir and a few others well known anti-covid-19 compounds. However, my suggestions to the authors were to validate their computational model with experimental data on some of their own identified Zinc database compounds. I leave it the editor for final decision for acceptance. 

Author Response

Thank you for your comment. Please note that the present manuscript is a review paper that has been created to evaluate and summarize the findings of existing studies. Herein, we report on computational articles of drugs that target (i) viral protease, (ii) Spike protein-ACE 2 interaction, (iii) RNA-dependent RNA polymerase, and (iv) other proteins and non-structural proteins of SARS-CoV-2. In this review article, compounds including Remdesivir, Lopinavir/Rotinavir, Sofosbuvir, Carvacol, and Favipiravir have been discussed. Moreover, the available data of clinical trials and experimental findings of the compounds reported from the reliable databases in this review. The authors will present the computational model with experimental data on identified Zinc database compounds in their future research article. 

This manuscript is a resubmission of an earlier submission. The following is a list of the peer review reports and author responses from that submission.

Round 1

Reviewer 1 Report

COMMENTS TO AUTHOR:

In this manuscript, the authors reviewed the computational view of repurposing drugs for Covid-19. However, the manuscript is not well-prepared and with minor errors. I would suggest the authors put more effort into improving the manuscript in terms of text and tables. While this manuscript may be suitable for publication in Biologics, there are a number of issues that need to be carefully addressed.

  1. In page 4, the author only used a small paragraph in introduction to discuss the current progress of computational method toward repurposing drugs for Covid-19. As the review paper is aimed for this topic, more contents should be discussed. Example, before the releasing of 3D crystal structure, what computational method could be used to determine the unknown protein structure (homology modeling) etc. More papers should be cited and addressed in this case. And for introduction, author should provide a brief background for terminologies like “docking”, etc.
  2. Table 1 should be more detailed comparing different results. And more work should be cited for comparison as well. In table 1, the author listed simulation time as one of the key parameters for comparison, however, all units are in ns, the comparison with such a small unit may not be much significant. Different force field may result in different binding affinity or docking score, therefore, I would recommend re-sort this table by force field, and put more key parameters into comparison, like glide score, docking score, glide emodel etc.
  3. Table 2 shares the same problem.
  4. For the future study, computational method could be a very promising tool toward drug discovery. In the discussion part, the author could try to compare those modeling results with some in vitro/vivo study data to highlight the significance of modeling in drug discovery.

Reviewer 2 Report

This review is devoted to clearly important topic – repurposing of existing drugs for treatment of COVID-19.

I think that the authors do not clearly understand the concept of existing works in this field.

The algorithm of most of studies is the following: docking of the existing drug compounds to the protein target, like main viral protease of SARS-CoV2, to find the leaders in docking energies to the requested site. Then molecular dynamics simulations of the best docked drugs to show stability of the complexes.

The review has to focus on docking strategies, the number of conformers of the compounds, methods of docking and generation of such conformers, etc. The MD simulations have to be clearly defined – how the protein-compound complexes were selected, what is the result of these simulations, RMSD, etc.

Simple creations of several tables is not a valuable result of the article.